# An Indirect Fluorescence Microscopy Method to Assess Vaginal Lactobacillus Concentrations

**DOI:** 10.3390/microorganisms12010114

**Published:** 2024-01-05

**Authors:** Ângela Lima, Christina A. Muzny, Nuno Cerca

**Affiliations:** 1Laboratory of Research in Biofilms Rosário Oliveira (LIBRO), Centre of Biological Engineering (CEB), University of Minho, Campus de Gualtar, 4710-057 Braga, Portugal; angela.martinslima@ceb.uminho.pt; 2Division of Infectious Diseases, University of Alabama at Birmingham, Birmingham, AL 35233, USA; cmuzny@uabmc.edu; 3LABBELS—Associate Laboratory, 4710-057 Braga, Portugal

**Keywords:** bacterial quantification, flow cytometry, fluorescence microscopy, colony forming units, vaginal lactobacilli

## Abstract

*Lactobacillus* species are the main colonizers of the vaginal microbiota in healthy women. Their absolute quantification by culture-based methods is limited due to their fastidious growth. Flow cytometry can quantify the bacterial concentration of these bacteria but requires the acquisition of expensive equipment. More affordable non-culturable methods, such as fluorescence microscopy, are hampered by the small size of the bacteria. Herein, we developed an indirect fluorescence microscopy method to determine vaginal lactobacilli concentration by determining the correlation between surface area bacterial measurement and initial concentration of an easily cultivable bacterium (*Escherichia coli*) and applying it to lactobacilli fluorescence microscopy counts. In addition, vaginal lactobacilli were quantified by colony-forming units and flow cytometry in order to compare these results with the indirect method results. The colony-forming-unit values were lower than the results obtained from the other two techniques, while flow cytometry and fluorescence microscopy results agreed. Thus, our developed method was able to accurately quantify vaginal lactobacilli.

## 1. Introduction

*Lactobacillus* species (spp.) frequently dominate the vaginal microbiota of healthy women [1]. The most common species of this genus present in the vaginal microbiota are *Lactobacillus crispatus*, *Lactobacillus jensenii*, *Lactobacillus gasseri*, and *Lactobacillus iners* [2]. Of note, *L. crispatus* is the most dominant vaginal lactobacillus in the majority of reproductive-age women [3]. Recent reviews on the role of *L. iners* in vaginal health have pointed out its presence in both optimal vaginal microbiota and in vaginal dysbiosis, making it unclear whether it is protective or pathogenic [4,5]. The dysbiotic state characterized by a depletion in vaginal lactobacilli with an increase in facultative and obligative anaerobes colonizing the vaginal milieu is designated bacterial vaginosis (BV) [6], the most common vaginal infection in women of childbearing age [7].

BV is clinically relevant as it affects millions of women annually and is associated with multiple complications [8,9] including pelvic inflammatory disease [10], preterm delivery [11], miscarriage [12], infertility [13], and an increased risk of acquisition of sexually transmitted infections (STIs) including HIV [14], gonorrhea, and chlamydia [15]. Furthermore, despite initial response to antimicrobial therapy, a large proportion of women with BV experience multiple episodes of recurrent disease [16]. However, after more than 60 years of intense research, the exact etiology of BV is still unknown [17]. A recent conceptual model for BV pathogenesis hypothesizes that virulent strains of *Gardnerella* spp. are responsible for the initiation of the BV biofilm, first by displacing lactobacilli from the vaginal epithelium, adhering to the vaginal epithelium in high numbers, and then by initiating BV biofilm formation [18]. Key interactions between *Gardnerella* spp. and *P. bivia* then promote the next step of BV biofilm formation. During this period *Gardnerella* spp. Acts in synergism with *Prevotella bivia*, leading to the production of vaginal sialidase which disrupts the mucin layer on the vaginal epithelium. This disruption then promotes the adherence of other BV-associated bacteria (BVAB) to the maturing BV biofilm [18]. The incorporation of other BV-associated species in the BV biofilm can induce *Gardnerella* spp. virulence factors [19,20], further promoting the development of BV.

*Lactobacillus* spp. grow fastidiously due to gene loss that occurs during the course of their evolution, which affects their metabolism and requires a rich nutrient media [21]. Furthermore, vaginal *Lactobacillus* spp. are anaerobes that tolerate oxygen only to a certain extent [22]. Their aerobic growth may stimulate oxidative stress and cellular damage in lactic acid bacteria due to the production of reactive oxygen species [23]. These growth requirements explain why *Lactobacillus* spp. are difficult to cultivate [24]. Therefore, culture-based methods are not adequate to accurately quantify lactobacilli concentration [25], which is essential for the preparation of an initial inoculum for downstream experiments [26]. Moreover, lactobacilli concentration quantification is important in the food industry, as *Lactobacillus* spp. are responsible for nutritious and organoleptic qualities of some food products [27,28,29]. In addition, lactobacilli quantification can be used in the detection of BV by microbiological and molecular methods [30] and to evaluate their role as probiotics [31,32].

Common molecular and non-molecular, culture-independent methods that allow for researchers to quantify bacteria include quantitative polymerase chain reaction (qPCR), flow cytometry (FC), and fluorescence microscopy (FM), respectively [33]. qPCR is widely use to quantify bacteria [34], due to its reproducibility, ease of experimental design, robustness, and affordability [35]. qPCR requires knowledge of the sequence data of the specific target gene of interest, but allows for very high specificity [35]. However, the adequate use of qPCR to quantify bacteria requires the use of exogenous controls to normalize the results, which is often ignored in many microbiology research studies [36]. The second mentioned method, FC, is fast, robust, and reproducible for determining total and viable bacterial counts, with the potential for automation; however, it is not cost effective [37,38]. FM is more affordable than qPCR and FC, but more time consuming [39,40,41,42], and requires the use of a Neubauer chamber or ocular grids installed in the microscope [43]. Smaller dimensions of bacteria present a limitation in the accuracy of this method, due to locating bacterial cells at different focal points [44]. Moreover, not all microscopes have ocular grids installed. We previously used FM to quantify vaginal lactobacilli and other vaginal bacterial species, by determining the surface colonization of these species [45,46]. However, surface colonization quantification has limited application and is unable to quantify bacterial suspension concentration in standard measurement units, such as colony-forming units (CFUs) per milliliter. Thus, we aimed to develop an indirect FM method for absolute quantification of vaginal lactobacilli in an attempt to overcome these limitations.

The need to develop a method to quantify lactobacilli emerged from the need to further study bacterial interactions between vaginal lactobacilli and BV-associated bacteria (BVAB) in BV pathogenesis research [18]. To achieve this goal, we used an easily cultivable bacterium, *Escherichia coli*, with well-defined bacterial density for a certain optical density (OD), to perform an indirect comparison. We were able to transform a three-dimensional quantification (CFU/mL) into a two-dimensional quantification (CFU/cm^2^), which allowed for us to quantify vaginal lactobacilli concentrations in suspensions.

## 2. Materials and Methods

### 2.1. Culture Conditions and Strains

*L. crispatus* CCUG 44128, *L. jensenii* CCUG 44492, *L. gasseri* CCUG 44075, and *L. iners* CCUG 38955 A were grown on Columbia blood agar base (CBA) (Oxoid, Basingstoke, UK) supplemented with 5% (*v*/*v*) defibrinated horse blood (Oxoid), at 37 °C under 10% CO_2_. *Escherichia coli* ATCC 25922 was also grown on CBA supplemented with 5% (*v*/*v*) defibrinated horse blood, but in an aerobic atmosphere at 37 °C. *Candida albicans* SC 5413 was grown on Sabouraud dextrose agar (SDA; Liofilchem, Waltham, MA, USA) plates and incubated for 24 h at 37 °C, as described by Fernandes et al. [47].

### 2.2. Colony-Forming Units Counting Method

Each lactobacillus biomass was resuspended in 1× phosphate-buffered saline (PBS) (composed by 8 g·L^−1^ of NaCl, 0.2 g·L^−1^ of KCl, 0.2 g·L^−1^ of KH_2_PO_4,_ and 1.15 g·L^−1^ of Na_2_HPO_4_, with pH 7.4 ± 0.05) and sonicated 3 times for 10 s at 40% with 10 s intervals between cycles. Each suspension OD_620nm_ was adjusted to 0.5 and serial dilutions ranging from 10^−1^ to 10^−6^ were performed in 0.9% (*w*/*v*) NaCl. Then, 10 μL of each dilution was spread into CBA plates and incubated for 48 h at 37 °C under 10% CO_2_. For *E. coli*, biomass was resuspended in 1× PBS and sonicated 10 s at 33% on ice. This suspension was adjusted to OD_620nm_ of 0.125, 0.25, and 0.5, which were then serial diluted from 10^−1^ to 10^−6^ and plated into CBA plates. The *E. coli* plates were then incubated at 37 °C for 18 h. These assays were performed with two replicates and repeated three times on separate days.

### 2.3. Flow Cytometry Quantification

Suspensions of the four vaginal lactobacilli of interest and *E. coli* were prepared as mentioned above, and their OD was adjusted to 0.5. Lactobacillus suspensions were stained using acridine orange (Alfa Aesar, Haverhill, MA, USA) 0.002 mg/mL for 15 min in the dark at room temperature. The *E. coli* suspension was stained using SYBR green (Invitrogen, Waltham, MA, USA) 1:10,000 for 15 min. Two different dyes were used in an attempt to obtain the higher percentage of stained bacteria. FC (Cytoflex, Beckman Coulter, Brea, CA, USA) counts were performed using the FITC filter block. The sampling rate was 10 µL/min, and the total number of events was set to 30,000 cells. Data were acquired and analyzed using CytExpert 2.4 software; multiparametric analyses were performed on forward and side scatter. To perform the manual gating, 1× PBS was analyzed in the FC before the bacterial suspensions, in order to define the area that corresponds to the events that are not cells. Since all bacteria were resuspended in 1× PBS, the particles present in this solution appeared in every bacteria analysis, for this reason, it is crucial to define what corresponds to 1× PBS events. All assays were repeated three times on separate days.

### 2.4. Visualization of Bacteria and Yeast in the Neubauer Chamber

A suspension of *L. crispatus* was prepared, as mentioned previously, and a suspension of *C. albicans* was prepared by resuspending its biomass in 1× PBS. The OD_620nm_ of both suspensions was adjusted to 0.5 and then diluted 10×. Next, 10 μL of Acridine orange 0.02 mg/mL was added to 20 μL of each suspension. After 5 min of incubation in the dark, 10 μL of the mixture were added to a Neubauer chamber, a lamella was placed on it, and the cells were visualized under the 40× objective of an epifluorescence microscope (Olympus BX51, Olympus, Lisbon, Portugal), capturing images with a screen resolution of 1360 × 1024 pixels with the Olympus Cellsens software 4.2.1 and using filters capable of detecting acridine orange (BP 470–490, FT500, LP 516).

### 2.5. Fluorescence Microscopy Quantification of Total Counts

Suspensions in PBS 1× of the four *Lactobacillus* spp. and *E. coli* were prepared, as described above. For *Lactobacillus* spp. the OD_620nm_ was adjusted to 0.5, and for *E. coli* it was adjusted to 0.125, 0.25, and 0.5. Next, a 1:500 dilution of each suspension was prepared and 20 μL of each dilution was spread on epoxy-coated microscope glass slides (Thermo Fisher Scientific, Waltham, MA, USA), and the slides were dried at 60 °C. Cells were then fixed by adding 100% (*v*/*v*) methanol to the slide for 20 min to, followed by 4% (*w*/*v*) paraformaldehyde (Thermo Fisher Scientific) for 10 min, and 50% (*v*/*v*) ethanol (Thermo Fisher Scientific) for 15 min [48]. After fixation, cells were covered with 20 μL of acridine orange (0.02 mg/mL) for 5 min. The slides were subsequently visualized under the 40× objective of an epifluorescence microscope, Olympus BX51 (Olympus, Lisbon, Portugal), using filters capable of detecting acridine orange (BP 470–490, FT500, LP 516). The number of bacterial cells were manually counted at the adequate dilution (<100 bacteria per field). We have previously determined that at least 13 images per sample are needed to have a high correlation between bacterial counts and bacterial concentration [48]. The surface area of each image was determined using the pixel area equivalent to the Neubauer chamber. In our optical system, each image was equivalent to 3.56 × 10^−4^ cm^2^. All assays were repeated three times on separate days.

### 2.6. Indirect Assessment of Bacterial Concentration by Measurement of Bacterial Surface Area

After performing the total counts for *E. coli*, a calibration curve was constructed, relating the *E. coli* CFUs (*x* axis) for each optical density and the total number of cells fixed to the microscopic glass surface (*y* axis). To determine the lactobacilli suspension concentration in CFU/mL, we performed the fixation of a lactobacilli suspension in a microscopic glass surface and quantified the total number of fixed cells following the same procedure as for *E. coli*. The obtained valued of fixed lactobacilli/cm^2^ was then applied in the calibration curve equation to convert the measured cells/cm^2^ by CFU/mL(i.e., we determined the lactobacilli concentrations using data (CFUs and total counts) from an unrelated bacterial species, namely, *E. coli*).

### 2.7. Statistical Analysis

The obtained results from FM, FC, and CFU for the four *Lactobacillus* species of interest were analyzed using the *t*-test paired samples with Microsoft Excel 2019. A *p*-value ≤ 0.05 was deemed statistically significant.

## 3. Results

### 3.1. Comparison of CFU and FC in the Quantification of Vaginal Lactobacilli

To assess the culturability of our vaginal *Lactobacillus* spp. of interest, we prepared suspensions of each of the 4 species, that were adjusted to obtain approximately 10^8^ CFU/mL, as described above. Then, we assessed the bacterial concentration of each suspension by two independent methods. As seen in Figure 1, both *L. crispatus* and *L. iners* had a significantly lower CFU quantification, in comparison with the FC analysis (for *L. iners* the difference was approximately 4 log).

### 3.2. Neubauer Chamber to Quantify Bacteria vs. Yeast

To highlight the importance of this novel indirect method, we started by demonstrating the limitation of the Neubauer chamber to quantify *Lactobacillus* spp. As can be seen in Figure 2, we observed that *L. crispatus* can be found at different focal points due to its small dimensions (in comparison with the depth of the Neubauer chamber), which could lead to miscalculation on the initial bacterial concentration. For comparison purposes, we also included the yeast *C. albicans*, whose cells are significantly larger and could be observed throughout the different depths of view.

### 3.3. Correlation of Surface Area Bacterial Measurement with Initial Bacterial Density

In order to overcome the above focal point limitation of the Neubauer chamber, we aimed to correlate the bacterial concentration contained in an equivalent volume used on the Neubauer chamber, after fixation on a regular microscopic slide. To validate this proof of principle, we selected an easily cultivable bacterial species, *E. coli*, that is known to be accurately quantified by CFUs [49]. Three different bacterial concentrations of *E. coli*, quantified by optical density, were used in this experiment. As shown in Figure 3, this method allowed for us to correlate the initial bacterial concentration (a 3D measurement), to a 2D bacterial measurement (surface area of the fixated suspension), thus allowing for us to use FM for bacterial culture concentration determination by eliminating the focal point limitation of the Neubauer chamber.

### 3.4. Quantification of Lactobacillus spp. Concentration by an Indirect Microscopy Method

By applying the same experimental procedure to the lactobacillus suspensions, we were able to infer their initial bacterial concentrations from the analysis of the quantified adhered bacteria, as shown in Figure 4. After performing total counts of the lactobacilli, we applied these results to the equation in Figure 3, converting total counts into CFUs. To confirm the accuracy of this method, we used the same bacterial concentrations used in the FM analysis and compared them to FC quantification. As can be seen in Figure 5, both methods resulted in very similar bacterial concentrations, despite some significative differences for *L. jensenii*, which had a lower concentration observed for the FC measurements.

## 4. Discussion

Bacterial quantification is a common and required experimental step in a wide variety of experimental set ups, including medical settings [50,51], food industry [52,53], environmental quality control [54,55], and industry [56,57]. Traditionally, the plate method colony counting is considered the gold standard for bacterial quantification [58,59]; however, with the realization that many bacterial species can develop viable but not cultivable state [60,61,62], non-culture-dependent methods have been developed [63,64]. However, development of new non-culture-dependent methods highlighted that often, different methodologies can yield significant different bacterial quantification [65,66,67]. As such, depending on the experimental design and model organism, accurate bacterial quantification might require the utilization of multiple techniques.

The results of this study demonstrate a feasible and affordable indirect method to determine vaginal *Lactobacillus* spp. concentrations. With the exception of *L. iners*, these species are considered as vaginal health markers, protecting the vaginal epithelium from colonization of pathogenic microorganisms [1]. However, due to their fastidious nature, these species are difficult to cultivate [24], as demonstrated in Figure 1. Previously, Lahtinen et al. compared four methods for the quantification of probiotic bifidobacteria and concluded that the choice of quantification method for fastidious bacteria may have a significant effect on the results of the analysis [68]. Thus, taking into consideration lactobacillus growth characteristics, we considered the use of culture-independent methods to enable their quantification, namely FC and FM. Our aim to develop an affordable, fast, and accurate method to quantify vaginal *Lactobacillus* spp. was motivated by the development and application of in vitro models to better study BV pathogenesis, in particular bacterial interactions between protective lactobacilli and key BV-associated bacteria (BVAB) [18]. We have previously proposed that some species of *Gardnerella*, the genus thought to initiate the formation of the polymicrobial BV biofilm, form a scaffold for which other BVAB can adhere, leading to a mature, polymicrobial biofilm. We have shown that *Gardnerella* spp. displace pre-adhered *L. crispatus*, which we hypothesized to be a pivotal event for BV development [69]. However, some Lactobacilli have also been shown to impair *Gardnerella* spp. [70,71] and other BVAB [72,73]. As such, it is important to better study interactions between Lactobacilli and BVAB, in order to better understand BV etiology, and the method developed herein can contribute to this goal.

Regarding the quantification of the four *Lactobacillus* spp. by CFUs and FC, the results of this study show that CFUs are not an appropriate method for accurate quantification of *L. crispatus* and *L. iners*, which is not surprising, as these species are considered difficult to cultivate [74]. Also, when comparing Figure 1 and Figure 5, we observed that the FC and FM results are more similar between all species (average difference of 0.73 ± 0.08 CFU/mL) than the results of FC and CFU (average difference of 1.31 ± 0.19), as well as FM and CFU (average difference of 1.81 ± 0.18), wherein CFU determination often underestimates the bacterial concentration. Moreover, other studies have also reported smaller bacterial concentrations determined by CFUs compared to FC and FM quantifications, suggesting that some of these bacteria may have become dormant [68,75,76].

We show the limitation of the Neubauer chamber to quantify bacteria, which we confirmed in Figure 2, with bacteria appearing in certain focal points, which is corroborated by the literature [44]. The reported lactobacilli and *C. albicans* dimensions found justify this limitation, with lactobacilli length as small as 2 µm [77], while *C. albicans* could have a diameter of approximately 12 µm [78]. Considering this and the high costs of FC [24], we developed a method based on FM to quantify vaginal lactobacilli, as this technique is readily available in many research labs [79]. We have shown that the relationship between total counts and plate counting for *E. coli* for three OD_620nm_ is linear, which allowed for us to determine the correlation between a surface area (2D) bacterial measurement with the initial bacterial concentration obtained with CFUs (3D). Thus, after counting total lactobacilli using FM, we converted those results to CFUs/mL.

Comparing the FC and FM results, statistical differences were verified only for *L. jensenii*; however, these differences were smaller than 1 log. Some differences between different methods are to be expected, and the subjectivity of cytometry manual gating (the process of identification of a cellular population) may have contributed to these differences [80,81]. A gate is a region in a biaxial plot of forward and side scatter, in which the cells with the desired phenotype are chosen. This first cell’s gate is then visualized in another biaxial plot, being redefined. However, as this is a manual process, it becomes subjective, as the sequence of biaxial plots depends on the user’s knowledge and view [82]. Nevertheless, for the other three vaginal lactobacilli there was agreement between quantifications obtained by FC and FM. Previous agreements between FC and FM measurements have been reported in studies with other microorganisms [83,84]. This supports the accuracy of our developed method. Precision, which is also important in fluorescence microscopy [85], was not determined in these experiments. However, imprecision can be overcome by counting more images [86]. We previously observed that a minimum of 13 images per sample would be required to obtain high precision, and no significant increase in precision was observed after increasing the number of images up to 20 images per field [48]. Our strategy to convert a surface area measurement in a bacterial concentration, using data from an unrelated species (*E. coli*) whose CFU relationship to OD is linear, was proven successful and can be used in future studies that require accurate assessment of vaginal lactobacilli concentrations. This method also has the potential to be adapted for other fastidious and/or dormant-inducing bacterial species including fastidious BV-associated bacteria.

## Figures and Tables

**Figure 1 microorganisms-12-00114-f001:**
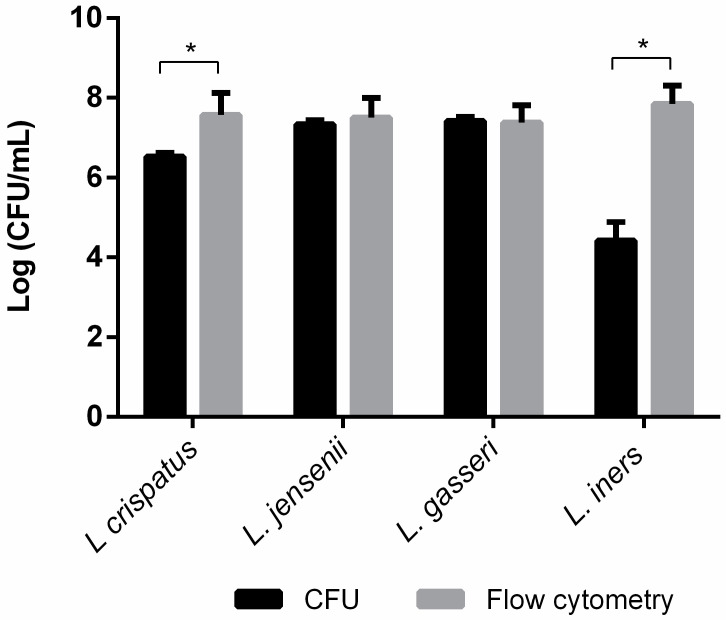
Quantification of bacterial concentrations of *L. crispatus*, *L. jensenii*, *L. gasseri*, and *L. iners* suspensions with an OD of 0.5 by culture-dependent (CFU) and -independent (Flow cytometry) techniques. The bars represent the mean and the standard deviation of 3 independent experiments. * Represents statistical significance (*t*-test paired samples, parametric, *p* ≤ 0.05).

**Figure 2 microorganisms-12-00114-f002:**
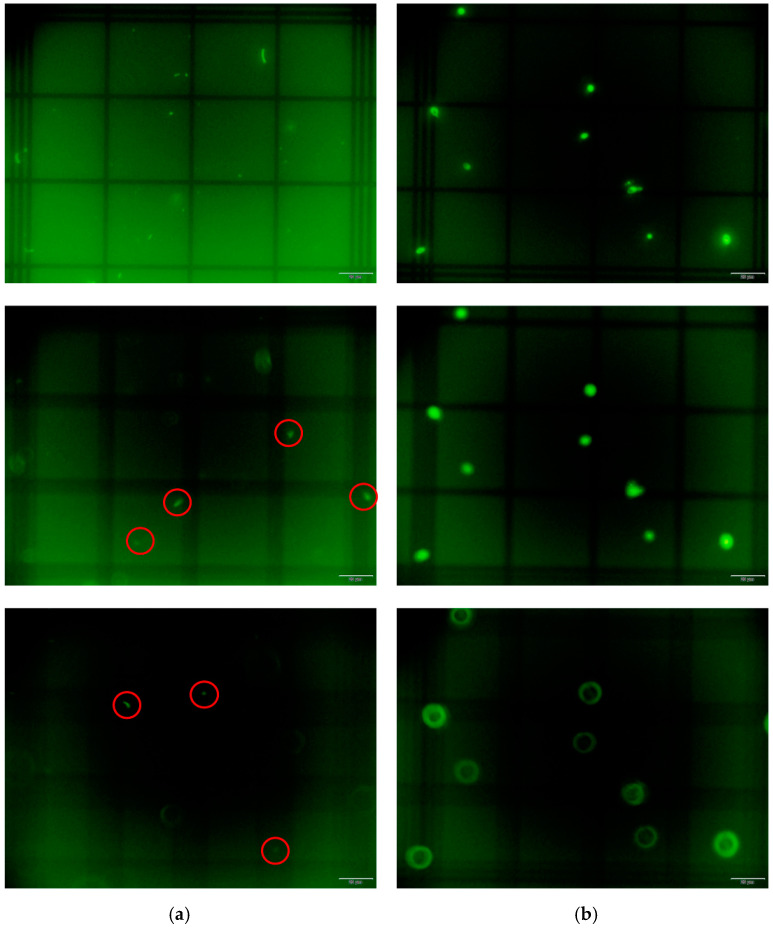
Comparison of the use of Neubauer chamber for (**a**) *L. crispatus* and (**b**) *C. albicans* using three z-stacks. The observation of these microorganisms was performed using fluorescence microscopy. For *L. crispatus* (**a**), the variation of focal points demonstrates different numbers of counts, while for *C. albicans* (**b**) the number of cells is the same for the three z-stacks. The red circles surround the bacteria that are not visible in the first focal plan. Scale bar: 20 µm.

**Figure 3 microorganisms-12-00114-f003:**
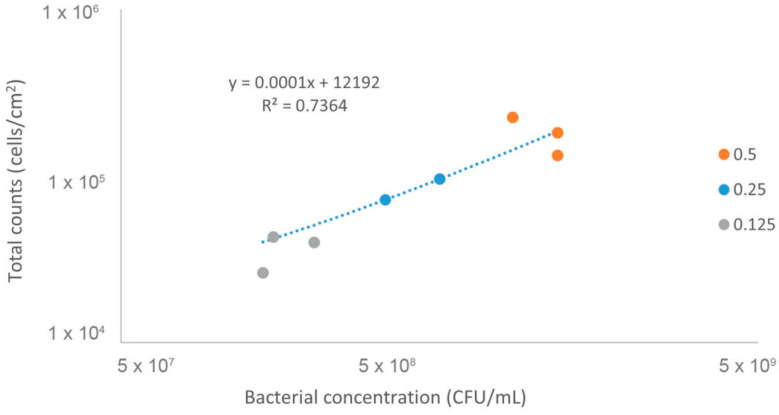
The relationship between *E. coli* CFUs and total cell counts using fluorescence microscopy for the same OD. Three bacterial suspensions initially adjusted to different OD_620nm_, namely, 0.5, 0.25, and 0.125, were quantified by both FM and by plate counting. The dots correspond to three independent assays for each OD.

**Figure 4 microorganisms-12-00114-f004:**
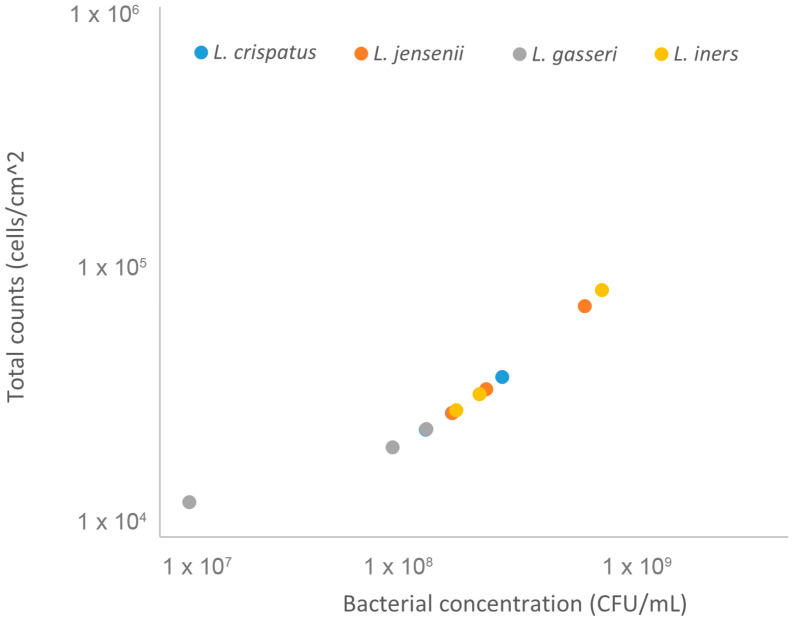
*Lactobacillus* spp. Concentrations obtained through total counts. Four bacterial suspensions adjusted to OD_620nm_ = 0.5 were quantified by fluorescence microscopy and the results obtained were converted into CFUs, as mentioned above. The dots correspond to three independent assays for each *Lactobacillus* spp.

**Figure 5 microorganisms-12-00114-f005:**
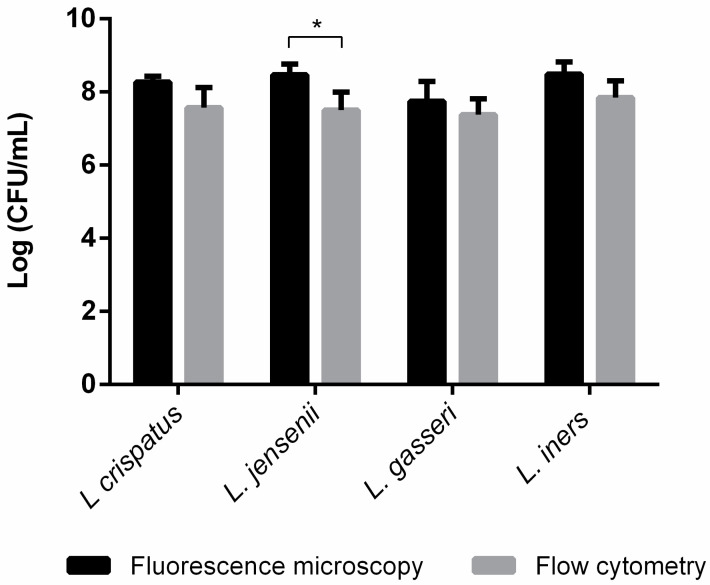
Quantification of bacterial concentrations of *L. crispatus*, *L. jensenii*, *L. gasseri*, and *L. iners* suspensions with an OD of 0.5, using fluorescence microscopy and flow cytometry techniques. The bars represent the mean and the standard deviation of 3 independent experiments. * Represents statistical significance (*t*-test paired samples, parametric, *p* ≤ 0.05).

## Data Availability

The data presented in this study are available upon written request to the corresponding author.

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
