# Peer review of "An Indirect Fluorescence Microscopy Method to Assess Vaginal Lactobacillus Concentrations"

_microorganisms, 2024, doi:10.3390/microorganisms12010114_

Round 1

Reviewer 1 Report

Comments and Suggestions for Authors

This is a straightforward, useful description of a technique that can be used to enumerate fastidious bacteria. The design for testing the technique was rigorous and well thought-out. A few minor points that should be addressed are listed below.

Title, Lines 21 and 132 and throughout: It is not clear why this is referred to as an “indirect method” or even as “indirect microscopy” as the bacteria are being directly observed and directly counted. The only thing indirect is the conversion of surface area to volume but conversion factors don’t make a technique indirect.

Lines 222-232: The Discussion would benefit from some mention about precision versus accuracy of this and other methods as precision is often a more important factor as it results in reproducibility. That, and some mention that for certain experiments, counting cultivable bacteria might be more important, desirable, and/or accurate. Accuracy in the case of this manuscript is measured only by the similarity to flow cytometry. But if an experiment calls for cultivable bacteria, then CFU would be more accurate. Additionally, there should be discussion about the error introduced during dilution, which affects the accuracy of estimation of the starting inoculum in all methods that require a dilution series.

Line 286: Why are the authors restricting this to Lactobacilli? Maybe an explanation about why lactobacilli were used as a model for testing the method and then a statement about why this would, or would not, be applicable to other bacteria.

Line 274: Are there any intrinsic characteristics of L. jensenii that might explain the discrepancy between flow cytometry and microscopy counts?

Author Response

our point by point answers are provided in the attached document

Reviewer 2 Report

Comments and Suggestions for Authors

This experimental study aimed at comparing three methods of quantitation of vaginal lactobacilli in suspensions. The study is clearly presented, methodologically sound, which makes the results reproducible. What I am missing is a mention of quantitative PCR, a robust, really affordable and accurate method for quantification of bacterial load in both cultures and clinical samples. The authors presented disadvantages of culture for quantification of fastidious bacteria, and, in my opinion, some reasoning regarding quantitative PCR would be relevant. Then, in the Discussion section, the authors stated that they aimed at developing an affordable, fast, and accurate method to quantify vaginal Lactobacillus spp., with a view of its utility in the development and application of in vitro models to study BV pathogenesis, and I would suggest mentioning this motivation in the Introduction too, to strengthen the background in terms of BV.  

 Minor comment

This paper was mentioned twice in the reference list (refs 2 and 3): Witkin, S.; Linhares, I. Why Do Lactobacilli Dominate the Human Vaginal Microbiota ? An Int. J. Obstet. Gynaecol. 2016, 124, 311 606–611, doi:10.1111/1471-0528.14390.

Author Response

Our replys to the reviewer comments are listed in the attached document
